# Precision Measurement System of High-Frequency Signal Based on Equivalent-Time Sampling

**Xiaoxuan Zang, Jianting Zhao \*, Yunfeng Lu and Qing He**

National Institute of Metrology, Beijing 100029, China; xxzang1005@163.com (X.Z.); luyf@nim.ac.cn (Y.L.); heqing@nim.ac.cn (Q.H.)
\* Correspondence: zhaojt@nim.ac.cn

**Abstract:** A high frequency periodic signal measurement system based on equivalent sampling method is developed. A high-speed sampling voltage tracking circuit, the core component of the system, is described in detail. The circuit can transform the amplitude corresponding to different phase points of the signal undertest into the equivalent DC level through successive approximation of multiple periods. The measurement system designed in this paper completes digital sampling with high accuracy only by connecting the low-cost voltage tracking circuit to the existing commercial instruments, such as two-channel waveform generator and high-precision digital multimeter, which makes the method easy to be generalized. The special structure of the sampling tracking circuit greatly reduces the influence of random noise and time jitter on the measurement results. The experimental results show that the non-linearity error of the system is as low as 0.002%, the bandwidth can reach 200 MHz, and the uncertainty of measuring the RMS of AC voltage with peak value of $\pm 1$ V and frequency of 10 kHz, 100 kHz and 1 MHz can reach $2.8 \times 10^{-4}$ V, $4.6 \times 10^{-4}$ V and $2.0 \times 10^{-4}$ V (k = 2), respectively.

**Keywords:** equivalent-time sampling; high-frequency voltage metering; voltage tracking circuit; high-resolution data acquisition

## 1. Introduction

Radio, navigation, radar and other fields are inseparable from the metering of high frequency voltage, which is of great significance to scientific research, production and national defense [1–3]. In the early stage, thermoelectric conversion and AC/DC substitution methods with stable performance and high accuracy were mainly used to measure RMS value of high-frequency signals. With the development of integrated circuit technology, digital acquisition and computer technologies, measuring various parameters of waveforms by digitizing the analog signals has become a popular research direction. In order to sample the high-speed analog signal without distortion, according to the Nyquist sampling theorem, the sampling frequency must be twice of the frequency of the signal undertest. In practice, in order to recover the time domain waveform accurately, the sampling frequency must be at least 10 times higher than frequency of the signal undertest, which means that high-speed sampling technology must be used. This puts forward high requirements for device speed, circuit structure and system debugging, thus increasing the difficulty of system implementation [4,5]. Although the sampling rate of ultra-high speed analog-to-digital converter (ADC) existing in the prior art can reach several or tens of GHz [6–9], the ADC chip still cannot meet the requirements of high resolution and high sampling rate for precision measurement at the same time because of the limitations of the integrated process. In order to improve the ability of the acquisition system to capture signals with the existing ADC technology, many scholars began to search the alternative sampling methods including time-interleaved sampling (TIS) theory [10] and multi-coset sampling (MCS) [11,12], which can both multiply the sampling rate but have high implementation

complexity. In addition, the differential sampling method can realize the measurement of AC voltage using quantum voltage as the standard signal [13], which significantly reduces the uncertainty of voltage amplitude measurement. The National Institute of Metrology, China (NIM) designed a precise differential sampling system in 2017 [14], which can measure signals at frequencies below 1 kHz, but it is difficult to measure signals at higher frequencies accurately.

It has been recognized that a broadband analog signal that needs to be collected is often periodic, and if the object to be tested is the reference signal, the signal also has high stability. Equivalent sampling can obtain data in different periods through multiple triggers and multiple sampling, and then recombine the sampled data to reconstruct the waveform of the original signal [15,16]. In theory, the equivalent sampling can recover the waveform of signal at a relatively low sampling frequency and the frequency of signal is much higher than the requirements of the Nyquist sampling theorem, so that high-speed data acquisition is able to be realized through the equivalent sampling method with using the periodicity of signal. Thus, the difficulty of system implementation is reduced, and the problems of high-speed data acquisition of periodic broadband analog signal are simplified.

The equivalent-time sampling method is widely used in digital oscilloscopes, such as the DSA8300 series equivalent-time sampling oscilloscopes launched by Tektronix and the 86,100 series sampling oscilloscopes launched by Keysight. They both have input bandwidth up to 80 GHz and equivalent sampling rate up to 10 TSPS. Although applying the equivalent sampling technique in oscilloscope can achieve a quite high sampling rate, the measurement accuracy of oscilloscope still cannot meet the requirement of precision measurement.

The National Institute of Standards and Technology (NIST) has carried out a series of studies on the waveform digitalization system based on equivalent sampling since the 1980s [17]. After more than 20 years of exploration and improvement, the system has reached the same accuracy as the AC-DC thermal transfer standard for AC voltage measurements over the frequency range of 10 Hz~1 MHz. In the frequency range from 1 kHz to 1 MHz, the sampler's gain flatness of the system is better than that available from the best commercial digital multimeter [18]. The system has a bandwidth of 2.3 GHz, excellent linearity and small time base jitter, which can accurately measure the RMS value of periodic high frequency signals from 10 Hz to 200 MHz. In the frequency range of 10 Hz~100 kHz, 100 kHz~1 MHz and 1 MHz~50 MHz, the relative uncertainty can reach $2 \times 10^{-5}$, $1 \times 10^{-4}$ and $2 \times 10^{-3}$, respectively. However, the realization of the sampling and digitizing system of NIST is a huge project, involving the design and implementation of a variety of complex functional circuits, such as fast pulse generator, special sampling comparator probe, precise time-based circuit and embedded development system, etc., which requires quite a long research time and extremely expensive implementation cost.

The digitizing measurement system based on equivalent time sampling described in this paper completes digital sampling with high accuracy only by designing and implementing a low-cost voltage tracking circuit on the basis of making full use of the external trigger and signal synchronization function of the existing commercial instruments, and the measurement repeatability achieves the accuracy of commercial standard source with the metering level. In addition, since the two-channel waveform generator and high-precision digital multimeter are commonly available in most laboratories, the high-precision sampling and measurement prototype system of high frequency periodic signal described in this paper can be readily reproduced.

The remaining sections are organized as follows: Section 2 introduces the overall architecture and working principle of the system and then describes the specific design and implementation of each module in detail. The error of system is analyzed, and the characteristic parameters of the system are measured in Section 3. The results of measuring the RMS of high frequency signals with the system designed in this paper are shown in Section 4; the accuracy and stability of the system are analyzed through the experimental data. The uncertainty of the system is evaluated in Section 5. Section 6 concludes the work.

## 2. Design of High Frequency Signal Measurement System

### 2.1. Principle of Equivalent Sampling

The basic principle of equivalent sampling is to convert the fast and repetitive signal with high frequency into a slow signal with low frequency through sequential sampling. The process of sequential equivalent sampling is shown in Figure 1.

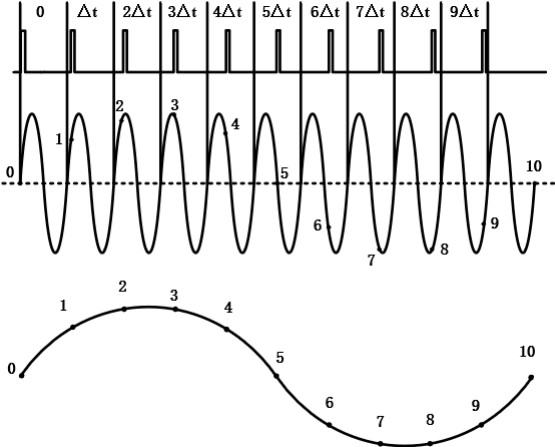

**Figure 1.** Schematic of sequential equivalent sampling.

Sampling begins after a very short time delay after each trigger event occurs, and the trigger delay adds a fixed interval to the preset sampling delay until all sampling points are obtained [19]. The time interval $\Delta t$ of each step determines the highest sampling rate $f_s$ of equivalent sampling.

$$f_s = \frac{1}{\Delta t} \tag{1}$$

### 2.2. The Structure of System

The structure block diagram of the high frequency signal measurement system is shown in Figure 2. The system can be divided into three parts: phase locking module, equivalent sampling voltage tracking circuit module and voltage measurement module.

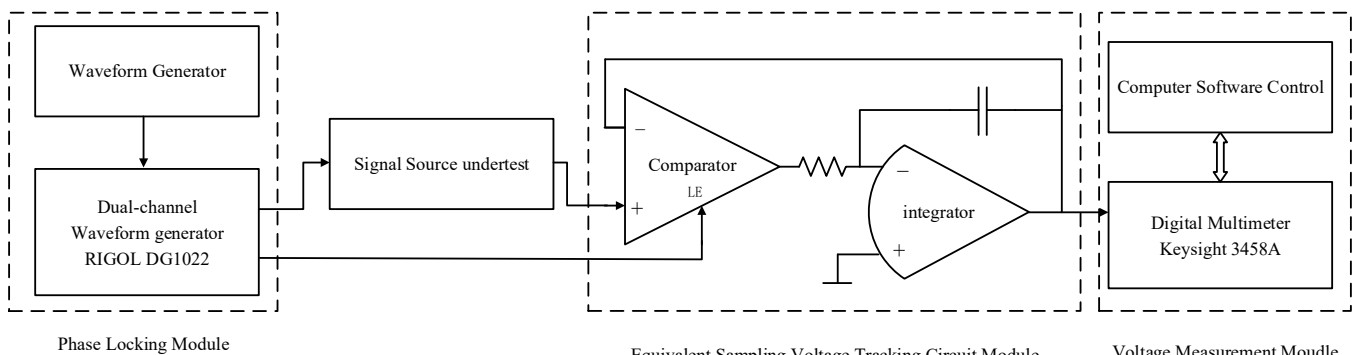

**Figure 2.** Structure of the system.

In the process of sampling digitization, the corresponding waveform of each signal is shown in Figure 3. The measurement process of the whole system is as follows: the waveform undertest is generated by the signal source undertest, and the strobe pulse is generated by the two-channel waveform generator in the phase locking module. The phase locking module completes the phase locking of the signal undertest and the strobe pulse through connecting and setting the instruments. The phase locking module is also able to adjust the phase difference between these two signals. Then, the equivalent sampling voltage tracking circuit transforms the amplitude of points on the waveform undertest

whose phase is corresponding to the strobe signal into the equivalent DC level after several periods of measuring and comparing (the specific principle of transform will be detailed in Section 2.2.2). The digital multimeter measures the value of the DC signal and sends the data to the computer, at which point the measurement of one sampling point is completed. The phase of the strobe pulse is then adjusted to change the corresponding points on the waveform undertest, and the sampling values at different phase are measured. The above process is repeated until the measured values of all sampling points are obtained. These sampling values come from different periods of the signal, and the data are reconstructed in one period, that is, the equivalent sampling of the high-frequency periodic signal is completed.

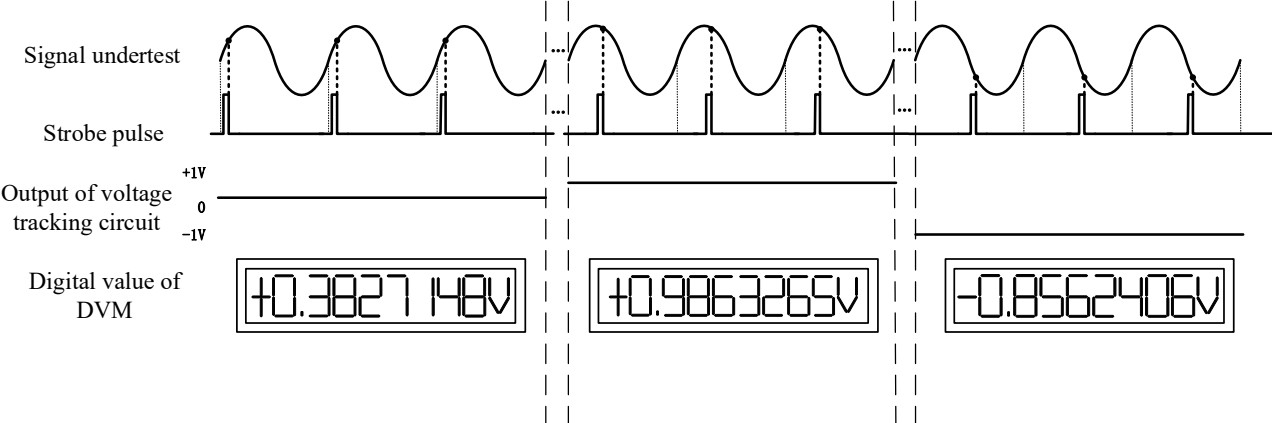

**Figure 3.** Waveform diagram illustrating the sampling digitization process.

The actual picture of the measurement system and its corresponding relationship with each module are shown in Figure 4. The specific design and principle of the three modules will be introduced in detail below.

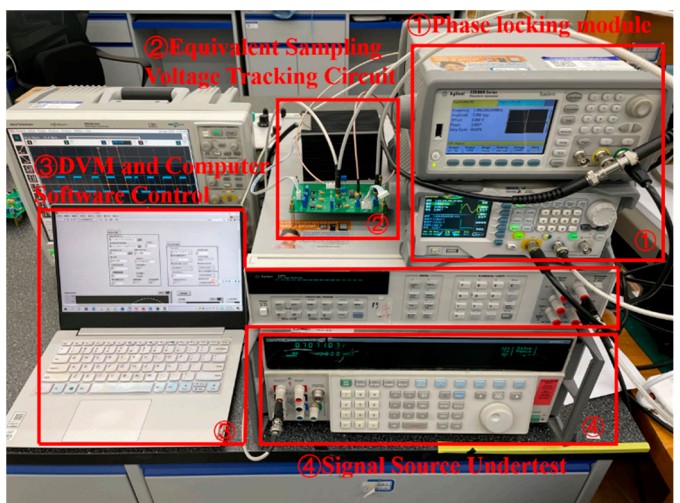

**Figure 4.** Physical drawing of measurement system.

### 2.2.1. Phase-Locking Module

In order to ensure the complete period sampling and the corresponding relationship between the sampling value and the phase, it is necessary to lock the phase of the strobe pulse and the signal undertest. A waveform generator with a dual channel output is selected, both channels are set to "Burst" trigger mode, and the same external trigger is input to the two channels through another waveform generator so that the signals output from these two channels are synchronized. One of the channels generates a narrow pulse

signal, which is output into the latch-enable terminal of the comparator as the strobe signal, and the other channel signal is output to the external-trigger terminal or phase-lock terminal of the signal source undertest to realize synchronization with the signal undertest. The relationship of three signals is shown in Figure 5. Then, phase locking of the strobe signal of comparator and the signal undertest is completed. At the same time, the phase difference of the above two signals can be adjusted by setting the start phase of the narrow pulse signal to "Burst" mode so as to realize the step of the sampling instant and the complete period equivalent sampling.

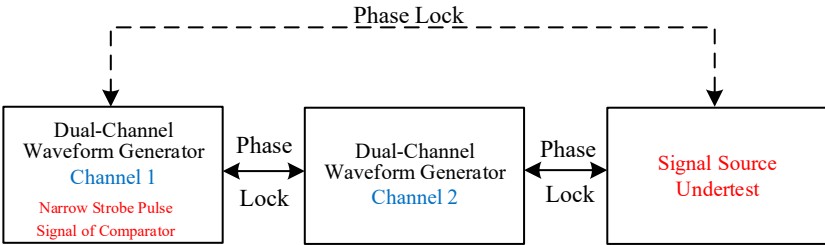

**Figure 5.** Phase locking correspondence.

RIGOL DG1022 dual-channel waveform generator is used in this paper. The frequency of its output function waveform can reach 25 MHz, the minimum pulse width of output pulse is 16 ns, and the adjustable range of phase is 0°~360° with a resolution of 0.001°. Therefore, the minimum stepping time interval of equivalent sampling can be 1/360,000 of the strobe pulse period. In other words, the equivalent sampling rate is able to be 360,000 times the frequency of the strobe pulse theoretically, which provides a guarantee for the wide frequency range of the signal that can be measured by the system of this paper. In practice, the maximum equivalent sampling rate of the system is also affected by the phase-locking accuracy of the signal undertest and the sampling pulse to a certain extent.

### 2.2.2. Equivalent Sampling Voltage-Tracking Circuit Module

The equivalent sampling voltage-tracking circuit is a loop generally composed of a comparator and an integrator. The circuit schematic diagram is shown in Figure 6

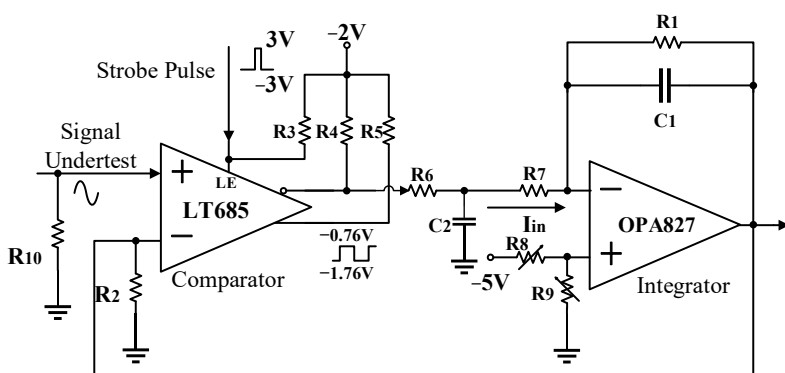

**Figure 6.** Equivalent-time sampling voltage tracking circuit.

The circuit uses the high-speed latch comparator chip LT685 as the sampler, which has short transmission delay, small input bias and high resolution, and is suitable for high speed and accurate analog-to-digital conversion processing. The comparator compares the signals of two input terminals when the latch-enable input terminal is receiving high level voltage and locks the current output state when receiving low level voltage.

Measuring the value of one sampling point on the signal undertest is taken as an example to illustrate the working principle of the equivalent sampling circuit. The waveform diagram corresponding to each signal is shown in Figure 7.

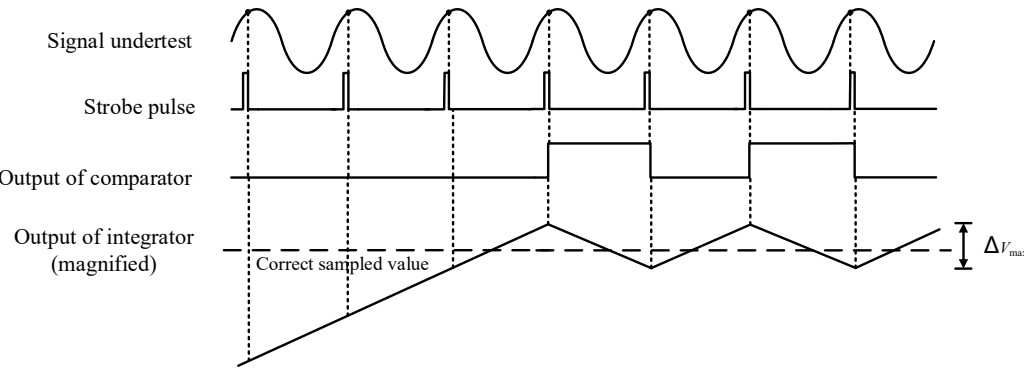

**Figure 7.** Corresponding waveform diagram of each signal.

When the level of the strobe pulse is high, the comparator compares the signal undertest at the non-inverting input terminal and feedback signal from integrator at the inverting input terminal; if the signal undertest is higher than the integrator output signal, the inverting output terminal of the comparator outputs a low level, which makes the integration capacitor connected to the comparator charge and the integrator performs forward integration, and vice versa. When the strobe pulse changes to a low level, the current output state of the comparator is latched, and the integrator performs forward or backward integration, whose duration is equal to the period of strobe pulse, so that the output of the integrator changes toward the value of the sampling point of the signal undertest. The strobe signal with a pulse width of 16 ns is periodic and synchronized with the signal undertest, so the sampling value of the signal undertest at this moment is compared with the output of the integrator repeatedly, and the circuit repeats the aforesaid process which makes the output of the integrator gradually approach the sampling value of the signal undertest at the strobe instant (as shown in top three periods of Figure 7), until the feedback of the integrated output signal is equal to the sampled value of the input waveform signal being tested, at which time the feedback signal oscillates about the sample value and the loop settles, which is shown in the last four periods of Figure 7. At this point, the transform of the sampling point is completed.

According to the measurement process and measurement principle described above, the correct sampled value will not necessarily equal the average value of the integrator output but may lie anywhere between the positive and negative limits as seen in Figure 7, thus, the maximum oscillation range of integrator output $\Delta V_{max}$ represents the resolution of digitization of a certain point on the waveform and it is important to select a small enough value for $\Delta V_{max}$. However, if the value is too small, as the output slope decreases, it may take an intolerably long time for the comparator–integrator loop to acquire a full scale step change. After fully balancing the measurement precision and measurement time, the maximum allowable error $\Delta V_{max}$ is set to 50 microvolts in this paper, which means the resolution of digitization of the system is 50 microvolts. If the frequency of the strobe signal is selected as 1 MHz, the relationship of the output slope and $\Delta V_{max}$ is as follows:

$$\frac{dV_{out}}{dt} = \frac{\Delta V_{max}}{T_0} = \frac{50 \ \mu V}{1 \ \mu s} \tag{2}$$

In Equation (2), $V_{out}$ is the output of the integrator and $T_0$ is the period of the strobe signal.

According to the circuit shown in Figure 6, the slope of the output can also be expressed by the input current of the integrator $I_{in}$ and integration capacitance $C_1$, which is shown as Formula (3).

$$\frac{I_{in}}{C_1} = \frac{U_i}{R_1 C_1} = \frac{dV_{out}}{dt} \tag{3}$$

In Equation (3), $U_i$ is the voltage difference between the positive and negative input terminals of the integrator, $R_1$ is the integration resistance. The integration constant will be as follows:

$$R_1 C_1 = \frac{U_i T_0}{\Delta V_{\max}} = \frac{U_i}{50 \text{ V/s}} = 10^{-2} \text{ s} \tag{4}$$

The value of $U_i$ is fixed as 0.5 V, so the integration constant $R_1 C_1$ can be calculated from the resolution of the system and the frequency of strobe pulse as shown in Formula (4). That is, the circuit can ensure the precision of the measurement by connecting resistance and capacitance with specific values.

By changing the phase difference between the strobe pulse and the signal being tested, the measurement of the next sampling point can be carried out, and the aforementioned process is repeated until the values of all the sampling points are obtained, that is, the equivalent sampling of the whole high-frequency periodic signal is completed.

This structure of voltage tracking circuit also has the following advantages:

1. Random noise in the system is averaged out by the integrator. The effectiveness of this noise averaging is determined by the integration constant and the number of samples taken at each time point;
2. The operational amplifier integrator operates at very low frequencies, essentially DC, so there is no need for a precision high-speed amplifier, which is a significant limitation in conventional sampling systems;
3. Enabling the latch comparator with narrow pulse can prevent the comparator from oscillating when the input differential voltage is very small, so that resolution of the measurement is not limited by such oscillation, and the sampling value can be tracked with higher accuracy.

### 2.2.3. Voltage Measurement Module

A Keysight 3458A Multimeter calibrated by quantum voltage is used to measure the DC voltage output by the equivalent sampling voltage tracking circuit in the voltage measurement module. The 3458A digital multimeter has excellent linearity, extremely low internal noise and high short-term stability. The relative uncertainty of measuring DC voltage in 24 h is less than $6 \times 10^{-6}$, which provides a strong guarantee for the accuracy of the system. In actual measurement, a LabVIEW program is written to control the sampling frequency, sampling points, integration time and other related parameters. The mean value of all samples obtained is taken as the final DC voltage measurement result, and the measurement data are stored at the same time.

## 3. System Characteristic Analysis

### 3.1. Non-Linearity Errors

The non-linearity errors of the system are mainly related to the comparator which can be measured as follows: A Fluke 5720A standard source calibrated by quantum voltage is used to input DC level of −1 V~1 V with interval of 50 mV to the system and a linear regression of the form of $Y = ax + b$ on the measurement results is performed. The obtained standard error of regression estimation is taken as the non-linearity error of the system. The estimated value slope and intercept can represent the actual gain and bias of the system respectively. The result of linear fitting of measurement data is as follows:

$$y = 0.9997x + 0.000031 \tag{5}$$

The results of fitting residual are shown in Figure 8, indicating that the maximum residual is less than 50 μV. The standard error of linear regression is 0.002%.

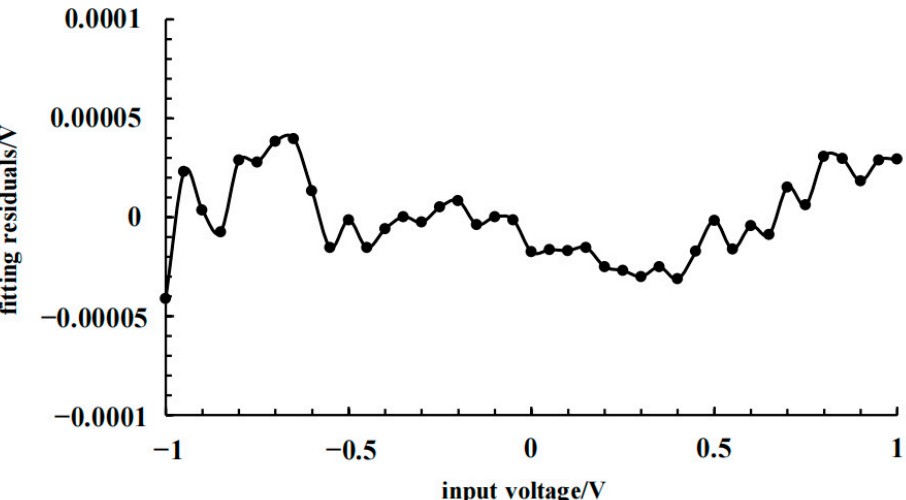

**Figure 8.** Fitting residual diagram.

*3.2. Time-Base Errors*

The time-base errors are mainly generated by the time-base jitter. Usually, the main reason for measurement error is that the slope of the measured signal is not symmetrical at both ends of the sampling instant. As a result, the absolute values of errors brought by the two-sampling jitters with the same absolute value, but different polarity are not equal, so they cannot offset each other which leads to the time-base error.

In the system described here, the synchronization of the sampling strobe pulse and the signal undertest is accomplished by triggering and phase locking between instruments. In practice, jitter exists in the internal time base of both the waveform generator and the signal source, which causes a relative phase error between the sampling pulse and the signal undertest. The influence of time-base jitter is analyzed by establishing a mathematical model for the measurement process of the system.

Suppose the signal undertest is $f(t)$ and its period is $T$, the output voltage of the equivalent sampling voltage tracking circuit is $V_D$. The measurement principle of the system is essentially to sample the signal undertest repeatedly in multiple periods at the particular moment $t$ when the sampling pulse corresponds to the signal waveform and compare the sampled value with $V_D$ at this time to determine the polarity of the output voltage change of the integrator, thus generating an output curve $V_D$ with random polarity and fixed slope. By sampling $V_D$ with a digital multimeter, a measurement data sequence is generated, and the estimated instantaneous value of the signal undertest $f(t)$ at moment $t$ is obtained by averaging this sequence.

Suppose the sampling value of the signal undertest is $f_t(k)$ and the value of $V_D$ is $V_{Dt}(k)$ during the $k$th sampling comparison at instant $t$. The time interval of sampling comparison at each point is $T'$, and $T'$ is an integer multiple of $T$. The circuit first compares the values of $f_t(k)$ and $V_{Dt}(k)$. If $f_t(k)$ is lower than $V_{Dt}(k)$, the comparator outputs high level, and the output of the integrator changes linearly in the direction of voltage reduction with a fixed slope. On the contrary, if $f_t(k)$ is equal to or higher than $V_{Dt}(k)$, the comparator outputs a low level, and the output of the integrator changes linearly with the opposite slope in the direction of voltage increase. According to Formula (3), the slope is $1/2RC$ and the absolute value $\delta$ of the change in $V_D$ between two adjacent samples is as follows:

$$\delta = \frac{T'}{2RC} \tag{6}$$

Thus, the expression of voltage $V_{Dt}(k+1)$ output by the integrator at the $(k+1)$th sampling comparison is

$$\begin{cases} V_{Dt}(k+1) = V_{Dt}(k) + \delta, V_{Dt}(k) \leq f_t(k) \\ V_{Dt}(k+1) = V_{Dt}(k) - \delta, V_{Dt}(k) > f_t(k) \end{cases} \tag{7}$$

A set of the measurement data sequence can be obtained through sampling $V_{Dt}(k)$, and the average value of this sequence can be used as the estimated value of $V_{Dt}$ of the signal undertest at instant $t$.

$$V_{Dt} = \frac{1}{n+1} \sum_{k=0}^{n} V_{Dt}(k) \tag{8}$$

Since the value of measured data sequence at the $(k+1)$th sampling comparison $V_{Dt}(k+1)$ is only related to the value at the $k$ sampling comparison $V_{Dt}(k)$, and has nothing to do with the value $V_{Dt}(i)$ $(i < k)$ before the $k$th sampling comparison, this sequence conforms to the definition of the Markov chain and has the properties of the Markov chain. The estimated value $V_{Dt}$ is called Markov estimation.

It can be seen from Formula (7) that the $(k+1)$th value $V_{Dt}(k+1)$ of the measured data Markov sequence is only related to the polarity of the value of $V_{Dt}(k) - f_t(k)$, and has nothing to do with the magnitude of this difference value. The absolute value $|V_{Dt}(k+1) - V_{Dt}(k)|$ of difference between two adjacent data in the sequence is constant, that is, the variation of two adjacent points in the Markov chain is a constant. Therefore, in the area of the jitter Markov chain is not sensitive to the slopes at both ends of the instant $t$, and the Markov estimate is also insensitive to the slopes at both ends of instant $t$.

The true value of the signal to be tested at moment $t$ is supposed to be $y_t$, and the median of the $f_t(k)$ sequence approaches $y_t$. In addition, since the Markov chain $V_{Dt}(k)$ is generated by comparing with $f_t(k)$ directly, the median of the sequence $V_{Dt}(k)$ is equal to the median of the $f_t(k)$ sequence, that is, the Markov estimate $V_{Dt}$ also approaches $y_t$. Therefore, the Markov estimate can be regarded as an unbiased estimate. Detailed and rigorous mathematical proof of the unbiasedness of Markov estimates has been carried out in ref. [20]. At the same time, it is mentioned in this paper that there is still a certain error with the maximum value of about 0.7% in Markov estimation near the extreme points (90° and 270°) of the signal under test. Therefore, the extreme points are avoided to be chosen during the measurement experiments.

To sum up, the measurement system described in this paper eliminates the system error introduced by time-base jitter in principle so that the time-base jitter error can be reduced to the required level of uncertainty by increasing the measurement time.

### 3.3. Bandwidth and Frequency Response

Although the calibrated high-frequency standard source cannot be obtained directly, the use of the commercial signal source to measure the frequency response of the system still has reference significance. In this experiment, Keysight E8257D high frequency analog signal generator with output frequency range of 250 kHz~20 GHz is used as the source to be tested, and the sinusoidal signals with peak value of ±1 V and frequency of 1 MHz~1 GHz are output to the measurement system.

The measured frequency response results of the system are shown in Figure 9. It can be seen from the figure that the corresponding frequency when the measurement amplitude attenuates to −3 dB is about 200 MHz, that is, the bandwidth of the measurement system is not less than 200 MHz.

Figure 10 shows the relationship between the gain error of the system and the frequency of the input signal. According to the diagram, the gain error of the system is less than 1% when the frequency of the signal undertest is in the range of 1 MHz to 70 MHz.

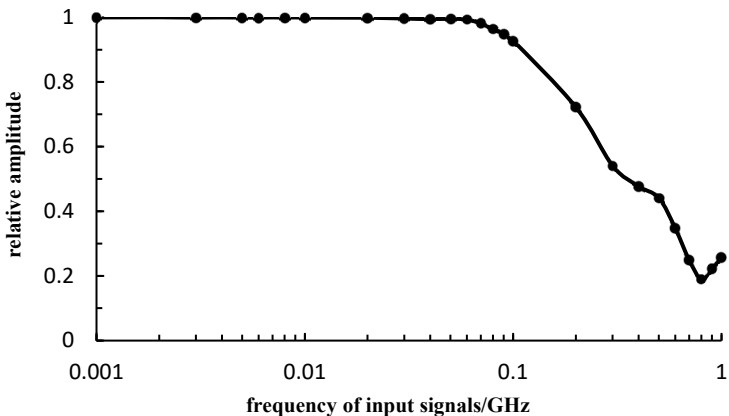

**Figure 9.** System frequency response.

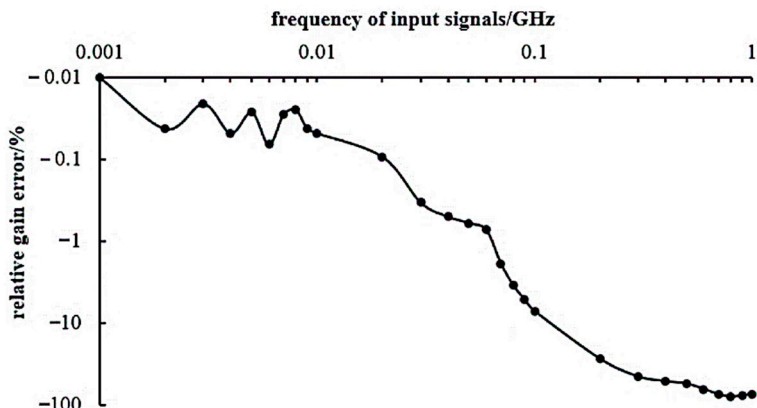

**Figure 10.** System relative gain error.

## 4. RMS Measurement Experiment

In order to verify the accuracy and stability of the measurement system, experiments are designed to measure the RMS of signals with different frequency, and the performance of the system is evaluated by the relative error and standard deviation obtained.

Since the output frequency of FLUKE 5720A standard source available is up to 1 MHz, only the accuracy of the system when measuring signals with frequencies below 1 MHz is evaluated. In the experiment, the FLUKE 5720A standard source is used to output sinusoidal signals with effective value of 0.707107 V and frequencies of 10 kHz, 100 kHz and 1 MHz. At range of 2.2 V, the 24-h relative uncertainty of 5720A at 10 kHz, 100 kHz and 1 MHz can reach $2.5 \times 10^{-5}$, $2.3 \times 10^{-4}$ and $1 \times 10^{-3}$, respectively. The RMS values of these three signals are measured by the system designed in this paper, respectively, and the relative errors and standard deviations of the three groups of data are analyzed. In addition, the RIGOL DG1022 signal generator is used to output sinusoidal signal with peak value of $\pm 1$ V and frequency of 10 MHz to the measurement system. Because the accuracy of waveform generator is unknown, only the standard deviation of this set of data is calculated to evaluate the repeatability of the system together with the above three sets of data.

When the frequency of the signal undertest is lower than 1 MHz, the frequency of the strobe pulse is the same as that of the signal undertest; when the frequency is higher than 1 MHz, the frequency of the strobe pulse remains unchanged at 1 MHz. The sampling point of the system is set to 24 and the integration time of 3458A digital multimeter is set to 0.2 s. Each sampling point is measured 100 times and the average value of this set of data is calculated as the estimated value of the amplitude of the sampling point. Taking the measurement of the aforementioned sinusoidal signal with frequency of 1 MHz as

an example, the data of 24 sampling points obtained and the reconstructed waveform are shown in Figure 11.

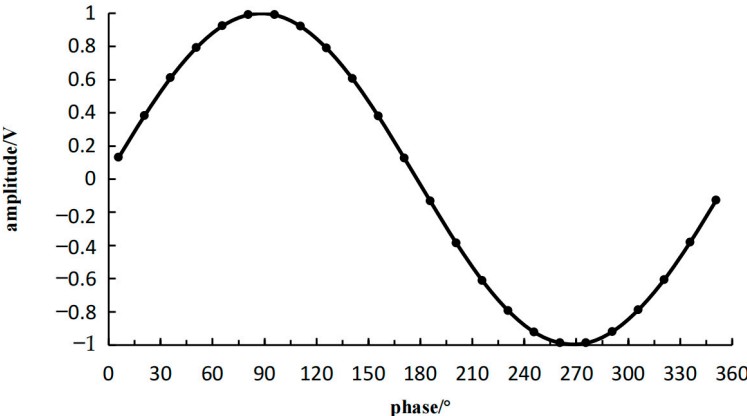

**Figure 11.** Measurement data and reconstructed waveform of 1 MHz sine signal.

The estimated value of the *i*th sampling point is denoted as $x_i$, and the RMS value can be calculated as Formula (9).

$$V_{rms} = \sqrt{\frac{\sum_{i=1}^{24} x_i^2}{24}} \tag{9}$$

It takes about 12 minutes to measure an RMS value and 12 RMS values are measured for each signal frequency A total of 288 sampling points of these 12 sets of data are not repeated, that is, the start phase of the first sampling point of each measurement of RMS is $0°$, $1°$, $11°$ and each sampling point is $15°$ apart. The measurement results of the RMS values of the signals with four different frequencies together with the average value and standard deviation are shown in Table 1.

**Table 1.** RMS measurement results/V.

| NO. | *f* | 10 kHz | 100 kHz | 1 MHz | 10 MHz |
|---|---|---|---|---|---|
| | 1 | 0.7070173 | 0.7068859 | 0.7061899 | 0.7068706 |
| | 2 | 0.7070022 | 0.7069289 | 0.7061609 | 0.7069150 |
| | 3 | 0.7069996 | 0.7069049 | 0.7061842 | 0.7069121 |
| | 4 | 0.7070518 | 0.7069544 | 0.7062047 | 0.7069099 |
| | 5 | 0.7069724 | 0.7069426 | 0.7061445 | 0.7068947 |
| Repeated | 6 | 0.7070517 | 0.7069473 | 0.7061826 | 0.7068896 |
| Measurements | 7 | 0.7070280 | 0.7069400 | 0.7062050 | 0.7068935 |
| | 8 | 0.7070244 | 0.7069241 | 0.7061826 | 0.7069222 |
| | 9 | 0.7070175 | 0.7069614 | 0.7061422 | 0.7068850 |
| | 10 | 0.7069914 | 0.7069488 | 0.7062014 | 0.7069119 |
| | 11 | 0.7070069 | 0.7068960 | 0.7061464 | 0.7070218 |
| | 12 | 0.7070180 | 0.7069275 | 0.7061421 | 0.7070433 |
| Average Value | | 0.7070151 | 0.7069301 | 0.7061739 | 0.7069225 |
| Standard Deviation | | $2.3 \times 10^{-5}$ | $2.4 \times 10^{-5}$ | $2.5 \times 10^{-5}$ | $5.4 \times 10^{-5}$ |

The relative errors of RMS values between the measurement data of the first three groups and the standard source are calculated, and the results are shown in Table 2. It can be seen from the Table 2 that the average relative errors of 12 groups of RMS values of 10 kHz, 100 kHz and 1 MHz signals are $-0.013\%$, $-0.025\%$ and $-0.13\%$, respectively, and the maximum relative errors do not exceed $-0.019\%$, $-0.031\%$ and $-0.14\%$. Because each RMS value are measured 12 times, the standard deviation of the average RMS values of

the 12 groups of 10 kHz, 100 kHz, 1 MHz and 10 MHz signals can reach $6.6 \times 10^{-6}$ V, $6.9 \times 10^{-6}$ V, $7.2 \times 10^{-6}$ V and $1.6 \times 10^{-5}$ V, respectively.

**Table 2.** Relative errors of RMS measurement values/%.

| NO. | *f* 10 kHz | 100 kHz | 1 MHz |
|---|---|---|---|
| 1 | −0.013 | −0.031 | −0.13 |
| 2 | −0.015 | −0.025 | −0.13 |
| 3 | −0.015 | −0.029 | −0.13 |
| 4 | −0.0078 | −0.022 | −0.13 |
| 5 | −0.019 | −0.023 | −0.14 |
| 6 | −0.0078 | −0.023 | −0.13 |
| 7 | −0.011 | −0.024 | −0.13 |
| 8 | −0.012 | −0.026 | −0.13 |
| 9 | −0.013 | −0.021 | −0.14 |
| 10 | −0.016 | −0.022 | −0.13 |
| 11 | −0.014 | −0.030 | −0.14 |
| 12 | −0.013 | −0.025 | −0.13 |
| Average Value | −0.013 | −0.025 | −0.13 |

## 5. Evaluation of Measurement Uncertainty

The evaluation of measurement uncertainty of system in this paper will be described below [21]. The uncertainty of the RMS value of AC voltage measured by the system with a peak value of ±1 V and a frequency of 1 MHz can be expressed as

$$V_{\text{rms}} = V_{ix} + \delta V_l + \delta V_A + \delta V_C \tag{10}$$

$V_{ix}$—RMS value measured by the system
$\delta V_l$—Influence of system nonlinearity on measurement results
$\delta V_A$—Influence of 3458A multimeter on the measurement result
$\delta V_C$—Influence of "thermal tails" errors of comparator on measurement results

- Measurement results of the RMS value, $V_{ix}$;

According to the repeated measurement of AC voltage with frequency of 1 MHz and RMS value of 0.707107 V, the standard deviation of sample is 25.2 μV. Because the measurements are repeated 12 times, the standard deviation of the average value of the 12 results is as follows:

$$u_1 = u(\overline{V}_{ix}) = \frac{25.2 \ \mu\text{V}}{\sqrt{12}} = 7.27 \ \mu\text{V} \tag{11}$$

- The linearity errors of the measurement system, $\delta V_l$;

The measured results show that the maximum linearity error of the system is less than 46.5 μV, which can be estimated by rectangular distribution. The introduced uncertainty component can be expressed as

$$u_2 = u(\delta V_l) = \frac{46.5 \ \mu\text{V}}{\sqrt{3}} = 26.8 \ \mu\text{V} \tag{12}$$

- Measurement uncertainty of 3458A multimeter, $\delta V_A$;

The 3458A digital multimeter is used in the system to measure the DC level. According to its user manual, the maximum 24 h uncertainty of the 1 V range is 1.8 μV, so

$$u_3 = u(\delta V_A) = 1.8 \ \mu\text{V} \tag{13}$$

- "Thermal tails" errors of the comparator, $\delta V_C$

The "thermal tails" errors of the comparator will affect the measurement results. According to the experimental results, the maximum error is not more than ±1 mV. It is

assumed that the error conforms to the two-point distribution in this range, and the introduced uncertainty component can be expressed as follows:

$$u_4 = u(\delta V_C) = 1 \text{ mV} \tag{14}$$

Table 3 shows the summary of uncertainty components:

**Table 3.** Summary table of uncertainty components.

| Source of Uncertainty | Symbol | Standard Uncertainty |
|---|---|---|
| Repeatability of measurement | $V_{ix}$ | 7.27 μV |
| Linearity errors of the system | $\delta V_l$ | 26.8 μV |
| 3458A multimeter | $\delta V_A$ | 1.8 μV |
| Thermal tails errors of comparator | $\delta V_C$ | 1000 μV |

According to Formula (10), the combined standard uncertainty of measuring RMS value of 1 MHz AC voltage by the system in this paper is $1.0 \times 10^{-3}$ V. When coverage factor k = 2, the expanded uncertainty is $2.0 \times 10^{-3}$ V.

$$u(V_{\text{rms}}) = \sqrt{u_1^2 + u_2^2 + u_3^2 + u_4^2} \tag{15}$$

Using the same evaluation method, the expanded uncertainty of measuring RMS value of AC voltage with frequency of 10 kHz and 100 kHz is $2.8 \times 10^{-4}$ V and $4.6 \times 10^{-4}$ V (k = 2), respectively.

## 6. Conclusions and Prospect

In this paper, a high frequency periodic signal measurement system based on equivalent sampling is designed. The instantaneous values of sampling points are converted into a corresponding DC level through successive approximation of multiple periods. The dynamic measurement of a high frequency signal is thus transformed into the static measurement. The high resolution of the phase adjustment of the two-channel waveform generator ensures the high equivalent sampling rate of the system. The designed structure of the sampling voltage tracking circuit and the setting of the integral constant guarantee the high resolution of the system. Experimental results show that the system can be used for accurate RMS voltage measurement of high frequency periodic signals from 1 MHz to 70 MHz; the gain error of the system is less than 1% in this region. The uncertainty of measuring the RMS value of AC voltage with peak value of $\pm 1$ V and frequency of 10 kHz, 100 kHz and 1 MHz can reach $2.8 \times 10^{-4}$ V, $4.6 \times 10^{-4}$ V and $2.0 \times 10^{-3}$ V, respectively (k = 2). The evaluation results also indicate that the uncertainty component introduced by the comparator thermal tail errors accounts for more than 98%. Therefore, further research can be devoted to reducing the thermal tail errors of the comparator to decrease the uncertainty of equivalent sampling measurement system to $10^{-5}$ level. In addition, the system demonstrated in this paper is mainly aimed at the measurement of the standard signal source with the synchronization function. In the future research, the front-end conditioning circuit can be added, and the synchronization can be completed by using the signal under test as the external trigger signal of the waveform generator that generates sampling pulse, thus, high-precision equivalent-time sampling can be achieved, even if the signal under test is output by an asynchronous signal source.

**Author Contributions:** Conceptualization, J.Z. Data curation, X.Z.; Funding acquisition, J.Z., Y.L. and Q.H.; Methodology, X.Z. and J.Z.; Project administration, J.Z., Y.L. and Q.H.; Software, X.Z.; Supervision, Q.H.; Validation, J.Z. and Y.L.; Writing—original draft, X.Z.; Writing—review and editing, X.Z. and J.Z. All authors have read and agreed to the published version of the manuscript.

**Funding:** This research was funded by National Key R&D program of China, grant number 2021YFF0603702.

**Conflicts of Interest:** The authors declare no conflict of interest.

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
