# Peer review of "Precision Measurement System of High-Frequency Signal Based on Equivalent-Time Sampling"

_electronics, doi:10.3390/electronics11132098_

Round 1

Reviewer 1 Report

A measurement approach has been proposed in this paper using the equivalent time-sampling. The overall contents do not provide a rigorous description regards with the state-of-the art contributions. A comparison results must be included to judge on the effectiveness of the proposed method and highlighting the advantages in the accuracy with respect to the complexity of the design set-up. Other comments to authors:

1.      How the precision is measured? It is better to include a clear formula for precision calculations.

2.      Both amplitude and phase noise have significant impact on the measurement accuracy. In addition, the noise increases in the high-sampling rate. It is better to discuss clearly such effects in the writing the paper.

3.      It is better to use white background for Figures 9 and 11 to be consistent with other figures.

Reviewer 2 Report

The authors propose a high sampling rate and high resolution data acquisition system that exploits a custom circuit and commercial instruments in an equivalent sampling approach. Overall the paper is clear and well structured, measured results are reported to validate the approach.
- It is not clear if the proposed approach can be applied also for cases where the signal to be measured is not a synchronized signal source. Alternative approaches could be discussed to cope with this issue (e.g. the signal is used as reference to synchronize pulses ?).
- Ref. [13] is not cited in the text.
- The authors could discuss more in detail the effect of jitter on strobe pulses
- What about the accuracy for higher frequencies (the highest testest frequency seems to be 10MHz)?
- What is k (=2) just before (10)? Please define

Reviewer 3 Report

ETS methods have been implemented in new oscilloscopes (for example: Keysight). Could you comment this?

According to the errors, the thermal tail error is very high and the presented other errors can be neglected due to theirs low value. Is it possible to reduce this error?

Reference no. 13 is not mentioned in the paper. Generally the references should be improved. Too few references from the last two or three years.

Round 2

Reviewer 1 Report

The authors addressed the suggested comments. However one minor suggestion is to include this reference an example of a precision measurement in frequency domain.

H. Al-Kanan, and F. Li, "A Simplified Accuracy Enhancement to the Saleh AM/AM Modeling and Linearization of Solid-State RF Power Amplifiers." Electronics 2020, 9, 1806.

Author Response

Dear Reviewer:

Thanks very much for your  comments concerning our manuscript entitled “Precision Measurement System of High Frequency Signal Based on Equivalent-Time Sampling” (ID: electronics-1753205). The main corrections in the manuscript and the responses to your comments are listed below.

-Comment : One minor suggestion is to include this reference an example of a precision measurement in frequency domain.

-Response :

We have made a modification and cited the article "A Simplified Accuracy Enhancement to the Saleh AM/AM Modeling and Linearization of Solid-State RF Power Amplifiers." H. Al-Kanan, and F. Li, Electronics 2020, 9, 1806. as reference 3 in the manuscript.

Reviewer 3 Report

Reference 19 is not cited.

Author Response

Dear Reviewer:

Thanks very much for your constructive comments concerning our manuscript entitled “Precision Measurement System of High Frequency Signal Based on Equivalent-Time Sampling” (ID: electronics-1753205). The main correction in the manuscript and the response to your comments are listed below.

-Comment : Reference 19 is not cited..

-Response :

We are very sorry for our negligence of omitting to indicate the position of reference 19 and we have made correction in the revised manuscript.